

# Uncertainty indication in soil function maps –
# Transparent and easy-to-use information to support sustainable use of soil resources

Lucie Greiner[1], Madlene Nussbaum[2], Andreas Papritz[3], Stephan Zimmermann[4], Andreas Gubler[1] ,
Adrienne Grêt-Regamey[5], Armin Keller[1]

[1] Swiss Soil Monitoring Network (NABO), Agroscope, 8046 Zürich, Switzerland
[2] School of Agricultural, Forest and Food Science (HAFL), Bern University of Applied Sciences (BFH), 3052 Zollikofen, Switzerland
[3] Institute of Biogeochemistry and Pollutant Dynamics, Swiss Federal Institute of Technology (ETH), 8092 Zürich,
Switzerland
[4] Forest Soils and Biogeochemistry, Soil Functions and Soil Protection, Swiss Federal Institute for Forest, Snow and Landscape Research (WSL), 8903 Birmensdorf, Switzerland
[5] Planning of Landscape and Urban Systems, Swiss Federal Institute of Technology (ETH), 8093 Zürich, Switzerland1

*Correspondence to*: Lucie Greiner (lucie.greiner@agroscope.admin.ch)

**Abstract.** The mapping of soil functions is increasingly being used to inform decision-making in spatial planning processes related to the capacity of soils to contribute to ecosystem services. In this study, we add to the transparency of soil function maps by indicating uncertainties arising from prediction uncertainties of soil properties as generated by digital soil mapping (DSM). For a study area in the Swiss Midlands, we map 10 static soil functions for agricultural soils together with their
uncertainties, using soil property data generated by DSM. Mapping the ten soil functions using simple ordinal assessment scales reveals pronounced spatial patterns with a high variability of soil function fulfillment (SFF) across the region, linked to the inherent properties of the soils and terrain attributes and climate conditions. Uncertainties in soil properties propagated through SFA methods generally lead to substantial uncertainty in the mapped soil functions. We propose two types of uncertainty maps that can be readily understood by stakeholders. Cumulative distribution functions of SFF scores indicate
that SFA methods respond differently to the propagated uncertainty of soil properties. Even where methods are comparable on the level of complexity and assessment scale, their comparability in view of uncertainty propagation might be different. We conclude that uncertainty indications in soil function maps are required to enable informed and transparent decisions on the sustainable use of soil resources.

**1    Introduction**

Human wellbeing relies on soil resources, and soil should therefore be better integrated into ecosystem service frameworks that inform decision-making and environmental policies (Dominati et al., 2010). Soil acts in multi-functional ways, and fulfils many functions in the regulation of the nutrient and water cycle, in carbon sequestration or the filtering of chemical





compounds, providing biodiversity and habitats for flora and fauna, and it is essential for the production of food, fibre and biomass (Adhikari and Hartemink, 2016; Haygarth and Ritz, 2009). The capacity of soils to deliver ecosystem services is largely determined by its functions, and each individual soil function can be seen as providing a soil-related contribution to ecosystem services (Bouma, 2014). The concept of soil functions has been increasingly been applied to reveal the role

played by soils in sustaining the wellbeing of humans and of society, emphasizing the multi-functionality of soils and their chemical, physical and biological properties. (Dominati et al., 2014; EC, 2006; Haygarth and Ritz, 2009; Makó et al., 2017; Schulte et al., 2014; Schwilch et al., 2016; Tóth et al., 2013). In general, soil function assessment (SFA) entails the rating of soils according to their capacity to fulfill an individual soil function, the so-called soil function fulfillment (SFF). Simplified static SFA methods result in scores that can be integrated into spatial planning procedures (Greiner et al., 2017). Maps that

enable visualization of SFF, so-called soil function maps, are well suited to communicating the importance of soils to spatial planners and other disciplines (Haslmayr et al., 2016; Sanchez et al., 2009) and can inform stakeholders on the role of soils for society and the environment (Bouma, 2010; Haygarth and Ritz, 2009; Miller, 2012).

In order to allow informed and transparent decision-making in spatial planning programs, however, balancing the social aspects of urbanization and environmental factors (Grêt-Regamey et al., 2017), not only must the state of soils with regard to

their functions be made available, but information on the reliability of the soil function maps is also required. Information on the accuracy of soil function maps facilitates decision-making for environmental policy, increases confidence among stakeholders, thereby helping to avoid poorly informed policy decisions with significant long-term environmental and social consequences (Maxim and van der Sluijs, 2011). At the same time, providing information on the uncertainty of soil function maps might delay decisions (Höllermann and Evers, 2017) or lead to discussions and negotiations in the spatial planning

process (Taylor et al., 2015). Nevertheless, the demand for soil information is considerable and stakeholders require not only the state of the soil in terms of soil quality, but also any indication of uncertainties associated with the soil information (Campbell et al., 2017).

Various sources of uncertainty can lead to spatially heterogeneous degrees of reliability in mapping soil functions. In general, the following types of uncertainties can be distinguished in assessing and mapping soil functions (Keller et al.,

2002): (i) model uncertainty that might arise from incomplete or incorrect methodological approaches and incomplete process descriptions, (ii) informational uncertainty of input data and model parameters, and (iii) temporal and spatial variation of soil properties. In the case of SFA, informational uncertainties in input data may result for instance from processing soil legacy data (Nussbaum et al., 2017a), prediction of soil properties using digital soil mapping approaches (DSM) (e.g., Nussbaum et al., 2017a; Sanchez et al., 2009; Vaysse and Lagacherie, 2015) or the application of pedotransfer

functions (PTF) (Chirico et al., 2010; Schaap, 2004) to deduce soil parameters from other soil properties.

We distinguish two SFA approaches that differ in their levels of complexity (Greiner et al., 2017). The static approach uses simplified empirical methods to assess the capacity of a soil to fulfil a specific function, neglecting the impacts of land use and land management practices. The static approach is particularly suitable for land-use planning to support the sustainable use of soil resources (Lehmann and Stahr, 2010). The dynamic approach takes into account soil processes and site-specific

environmental factors, as well as land use and land management practices. Dynamic models exist for nutrient and water cycling, carbon sequestration, crop production, and other soil functions (Vereecken et al., 2016). The use of dynamic soil models is both data-demanding and time-consuming, but is a powerful means of modelling the impacts of past and future land use and land management practices on soil functions. The assessment of uncertainties in environmental (dynamic)

modelling has been demonstrated in numerous studies (Bastin et al., 2013; Brown et al., 2005; Heuvelink et al., 2007, 2010; Krayer von Krauss et al., 2005; Lesschen et al., 2007) and various frameworks have been proposed to take into account sources of uncertainty (Bastin et al., 2013; Heuvelink et al., 2007). In contrast, uncertainties among static SFA approaches have hardly been accounted for at all.

In this study, we propagate prediction uncertainties in soil properties (informational uncertainty) through the calculation of

ten static SFAs for a case study area in the Swiss Plateau. The SFA methods used are presented in (Greiner et al., 2017) and were chosen to reveal the breadth of multi-functionality of soils. We used soil property maps generated using a digital soil mapping approach (DSM) that exploits soil legacy data (Nussbaum et al., 2017b) and has the advantage that the prediction intervals for soil properties are provided. The objectives of our study were to propagate soil property predictions through static SFA, in order to indicate how accurate the SFA results are in response to informational uncertainty and spatial

variation of soil properties as quantified by the DSM approach, and to gauge how sensitive the SFA methods are to predictive distribution in soil properties.

**2    Materials and Methods**

**2.1    Study area**

Our study area is located in the Swiss Plateau in the Canton of Zürich around Lake Greifensee, see Figure 3. The region is dominated by urban areas and agricultural land (crop production, mixed and dairy farming). We only assessed soils under

agricultural use. Urban areas, forest, wetlands, parks, and city gardens are excluded from this study, resulting in a total study area of 170 km$^2$. Fluvisols, Luvisols, Cambisols, Regosols, Gleysols or Histosols have developed in a versatile geology, but in general on quaternary molasses and moraines. The region lies at about 390-840 metres above sea level, and the vegetation time amounts to approximately 190 days per year. Slopes greater than 35% can only be found alongside moraines, otherwise the slopes are between 10 and 15% (Jäggli et al., 1998). More details on the region, its soils and its extent are provided in

(Jäggli et al., 1998; Nussbaum et al., 2017b).



**Figure 1 Study area in the Swiss Midlands. (Orthophotos study area: SWISSIMAGE 2005, ©SWISSTOPO. Administrative boundaries Europe: NUTS 2010, ©EuroGeographics)**

5 **2.2    Soil function assessment**

We assessed regulation, habitat and production functions for 10 soil (sub)-functions (Table 1) as proposed in a previous review by Greiner et al (2017). Each SFA method addresses a certain domain of the soils multi-functionality depicting a specific assessment criterion, e.g., the nutrient storage capacity of soils for the nutrient cycle. The SFA methods require data on soil properties, PTFs, and other environmental data (Table 1).

**2.2.1    Regulation functions**

We assessed the regulation of the water cycle (R-water) following the method proposed by (Danner et al., 2003), which





combines the water storage capacity (WSC in mm/m$^2$) of soils with their saturated hydraulic conductivity (SHC in cm/day) for a reference soil depth down to 1m. The nutrient storage capacity (NSC in mol$_c$/m$^2$) of soil is one of its most important parameters, determining the nutrient cycle (R-nutric). We calculated the NSC according to (Lehmann et al., 2013), multiplying the fine earth fraction (mass of clay and silt) and the amount of soil organic matter for each soil layer with its

effective cation exchange capacity (CEC$_{eff}$) down to a soil depth of 1 m. The method proposed by (Jäggli et al., 1998) evaluates the capacity of soils to prevent the loss of soil nutrients by runoff and percolation to ground and surface water (R-nutril). The SFA method takes into account basic soil properties as well as the hydromorphic properties of soils (waterlogging) and environmental site conditions. The capacity of the soil to filter and buffer trace metals (R-icont) were assessed for cadmium, copper and zinc using a method developed by the German Association of Water, Wastewater and

Waste (DVWK, 1988) to prevent groundwater pollution by trace elements. The SFA method evaluates the filtering capacity of topsoils (0- 30 cm) to retain trace metal cations based on sorption sites of organic matter, clay minerals, and sesquioxides in conjunction with soil pH and redox potential (DVWK, 1988).

The regulation of organic compounds (R-ocont) is assessed using the method of Litz (1998) for four frequently used herbicides in Switzerland: glyphosate, pendimethalin, metamitron and isoproturon (Franzen et al., 2017). The SFA method

assesses the potential sorption and fixation of an organic compound on clay and organic material (binding) and the potential biological activity of a soil to decompose an organic compound (decomposition). In a second step, both assessment criteria are combined to evaluate the retention potential of a soil for a specific chemical compound (retention). To account for the ability of soils to buffer acids (R-acid), we applied the SFA method proposed by (Bechler and Toth, 2010). The method takes into account the amount of clay and organic matter down to a soil depth of 1 m, and soil pH. To address the role of soils in

the carbon cycle (R-carbon) we simply calculated the soil carbon stock to 1m depth.

### 2.2.2    Habitat and production functions

We used the method proposed by (Siemer et al., 2014) to assess the capacity of soils to provide niches for rare plant species (H-plant). This is applied to sites with extreme soil properties and shallow soils that lead to relatively dry or wet soil

conditions or low nutrient availabilities, which provide niches for rare plant species. As an indicator of the habitat function we estimate soil biological activity based on empirical regression functions to estimate microbial biomass in grassland and arable soils (H-micoorg) (Oberholzer and Scheid, 2007). These PTFs were derived for hundreds of grassland and arable sites across Switzerland.

We assessed the agricultural production function (P-agri) using the method of Jäggli et al. (1998). This SFA method

combines basic soil properties, climate data (climate suitability classes depending on temperature, precipitation and length of growing period (BLW, 2012)), and site conditions (slope, topography) to classify soils into 10 classes according to their suitability for crop growth.





The results of SFA methods are usually given in physical or chemical units and transformed to an ordinal scale, i.e., an SFF score, to facilitate the communication of multi-functionality to stakeholders. In agreement with other studies assessing soil function (e.g., Miller 2012, Haslmayr et al. 2016, Lehmann and Stahr 2010), we applied an ordinal scale with five levels: SFF score = 1 (very low/very poor), SFF = 2 (low/poor); SFF=3 (medium), SFF= 4 (high/rich) and SFF=5 (= very high/very

5  rich).

**Table 1. The ten assessed soil functions for the case study area, their assessment criteria and required input data. For the uncertainty assessment soil properties were treated as fixed (SPm) or as random variables (SPd) (see Chapter 2.4 for explanation). SOM: soil organic matter, SC: stone content, WH: presence or absence of waterlogged horizons, DC: Drainage Class, AAC:**
10  **available air capacity in mm, AWC: available water capacity, BD: bulk density, CECpot and CECeff: potential and effective cation exchange capacity, MB: microbial biomass, SHC: saturated hydraulic conductivity, S-value: amount of exchangeably bound basic cations, type of method (see section 2.4. and Figure 2). * SOM for 50-100 cm depth: SPm**

| Soil function | Assessment criterion | Source of method | SP_d (Clay, SOM*, SC, pH) | | | | SP_m (Silt, Depth, WH, DC) | | | | PTF | Other environmental data | Type of method | Acronym |
|---|---|---|---|---|---|---|---|---|---|---|---|---|---|---|
| Soil (sub-)function | | | | | | | | | | | | | | |
| **Regulation function** | | | | | | | | | | | | | | |
| Water cycle | Water infiltration (cm/d) and storage capacity (mm/m²) combined in semi-quantitative look-up table | Danner et al. (2003) | x | x | x | | x | x | x | | BD, SHC, AWC, AAC | Slope, geology, climate | 2 | R-water |
| Nutrient cycle | Nutrient storage capacity of fine earth down to 1 m soil depth (mol_c/m²) | Lehmann et al. (2013) | x | x | x | x | x | x | x | | BD, CECeff | | 1 | R-nutric |
| Nutrient losses | Retention capacity against nutrient losses, e.g., nitrate (semi-quantitative look-up tables) | Jäggli et al. (1998) | x | x | x | | x | x | x | x | BD | Slope, geology, climate | 2 | R-nutril |
| Heavy metals | Sorption capacity for inorganic pollutants (semi-quantitative look-up tables) | DVWK (1988) | x | x | x | x | | x | | | BD | | 2 | R-icont |
| Organic compounds | Retention capacity for organic contaminants against percolation into ground water (semi-quantitative look-up tables) | Litz (1998) | x | x | x | x | x | x | x | x | BD, AWC, CECpot, S-value | Properties organic compounds, mean annual temperature and evaportranspiration, climate | 2 | R-ocont |



| | | | | | | | | |
|---|---|---|---|---|---|---|---|---|
| Acids and contaminants | Buffering and binding capacity for acids and contaminants assessed by soil organic matter content (in kg/m$^2$, clay content (in kg/m$^2$) and maximum pH in assessment depth combined in a semi-quantitative look-up table | Bechler and Toth (2010) | x x x x | x x | BD | | 2 | R-acid |
| Carbon cycle | Amount of organic matter pool in soil (C-storage) (kg C/m2) | Greiner et al. (2018) | x x | x | BD | | 1 | R-carbon |
| **Habitat function** | | | | | | | | |
| Plants | Soils providing niches for plant species, with very dry, wet or low nutrient properties (assessed by available water capacity in mm, presence of hydromorphic horizon and effective cation exchange capacity in cmol$_c$/kg) | Siemer et al. (2014) | x x · x x x x | | BD, AWC | | 2 | H-plant |
| Micro-organisms | Amount of microbial biomass (mg/kg dried soil) | Greiner et al. (2018) | x x · x x x | | MB | Land use | 1 | H-microorg |
| **Production function** | | | | | | | | |
| Agricultural production | Suitability for agricultural production (semi-quantitative look up tables) | Jäggli et al. (1998) | x x x x x x x x | | BD | Relief, slope, climate | 2 | P-agri |

## 2.3 Soil property maps and other data

Soil property maps were generated using the digital soil mapping (DSM) approaches of Nussbaum et al. (2017b) for the case

5 study area with a spatial resolution of 20 m raster cells. This resulted in a total of about 450 000 raster cells for the agricultural soils. In the DSM approach Nussbaum et al. (2017b) used a new boosted geoadditive modelling framework (geoGAM) in which they modelled nonlinear relationships and selected parsimonious models from a large number of covariates. Table 2 presents summary statistics of the modelled soil properties in our case study for the four soil layers that were distinguished. The accuracy of the predictions, validated using independent data, was similar to other DSM studies.

10 Independent models were fitted for each soil property and each soil depth (Nussbaum et al. 2017).

In order to apply the SFA methods, PTFs suitable for diverse soil parameters are required (see Table 1). To estimate soil bulk density we used the PTF of Nussbaum and Papritz (2015), and for the cation exchange capacity we used the PTF of Gerber (2014). Both PTFs were developed for Swiss soils based on soil legacy data. Available water capacity (AWC) and other soil hydraulic properties were estimated using the German soil mapping guidelines (KA5, 2005). Other environmental





data such as slope, relief, climate, geology, geomorphology, properties of organic compounds, and land use were gathered from available databases (BFS, 2010; BLW, 2012; HADES, 2017; PPDB, 2017; Swisstopo, 2008, 2014).

**Table 2. Summary statistics of modelled soil properties generated by the DSM approach by Nussbaum et al. (2017) for the Greifensee study area.**

| | Soil property | Depths | Mean | | | STD | | | Distribution |
|---|---|---|---|---|---|---|---|---|---|
| | | | Q0.1 | Q0.5 | Q0.9 | Q0.1 | Q0.5 | Q0.9 | |
| SP$_d$ | Clay (%) | 0-10 | 19.4 | 24.3 | 29.4 | 5.5 | 5.7 | 5.8 | - |
| | | 10-30 | 20.4 | 25.6 | 31.2 | 5.5 | 5.7 | 5.8 | - |
| | | 30-50 | 20.4 | 25.4 | 31.2 | 6.6 | 6.8 | 7.0 | - |
| | | 50-100 | 18.9 | 24.7 | 30.3 | 7.3 | 7.5 | 7.7 | - |
| | Soil organic matter (%) | 0-10 | 4.4 | 5.8 | 8.2 | 1.7 | 2.2 | 3.1 | - |
| | | 10-30 | 4.3 | 5.8 | 8.5 | 1.9 | 2.5 | 3.7 | - |
| | | 30-50 | 1.7 | 5.9 | 10.7 | 6.7 | 15.5 | 22.2 | - |
| | Stone content (%) | 0-10 | 3.1 | 7.6 | 12.6 | 3.5 | 5.8 | 7.5 | - |
| | | 10-30 | 3.4 | 8.3 | 13.7 | 3.7 | 6.0 | 7.9 | - |
| | | 30-50 | 4.0 | 9.9 | 18.1 | 4.6 | 7.7 | 10.5 | - |
| | | 50-100 | 5.4 | 12.6 | 21.2 | 6.4 | 10.2 | 13.5 | - |
| | pH | 0-10 | 6.2 | 6.5 | 7.0 | 0.5 | 0.5 | 0.5 | - |
| | | 10-30 | 6.1 | 6.5 | 6.9 | 0.5 | 0.5 | 0.5 | - |
| | | 30-50 | 6.1 | 6.5 | 7.0 | 0.6 | 0.6 | 0.6 | - |
| | | 50-100 | 6.2 | 6.6 | 7.0 | 0.6 | 0.6 | 0.6 | - |
| SP$_m$ | Soil organic matter (%) | 50-100 | | 1.0 | | | 0 | | - |
| | Silt (%) | 0-10 | | 34.8 | | | 2.2 | | - |
| | | 10-30 | | 35.5 | | | 2.3 | | - |
| | | 30-50 | | 32.9 | | | 3 | | - |
| | | 50-100 | | 33.6 | | | 3.1 | | - |
| | Soil depth (cm) | - | | 70.1 | | | 14.6 | | - |

### 2.4 Indication of uncertainty in mapping soil functions

In this study, we propagated uncertainties for four basic soil properties, i.e., clay content, SOM, pH and stone content, through the calculation of the ten static SFA methods. These four soil properties were treated in the calculations as random variables for each raster cell and soil depths 0-10 cm, 10-30 cm, 30-50 cm and 50-100 cm (Table 1). For the soil depth of 50-100 cm, SOM was treated as a fixed input variable *(SP$_m$)* because its predictive performance was too low (Nussbaum et al. 2017). For this depth we used the median of the available soil data (n = 418). The probability distributions of these soil properties (SP$_d$) were derived from the DSM approach mentioned above, performing 1000 simulations for each raster cell and soil depth (Nussbaum et al. 2017). For the calculation of the SFA we drew an independent set of the four SP$_d$ values



(drawn and replaced) N=1000 times, and compared range, mean and variance of the generated $SP_d$ set with the original distributions of the four soil properties predicted using the DSM approach.

We restricted the number of random variables to these four soil properties due to the required computation time for such a large number of raster cells with four soil depths. Therefore, other soil properties such as silt content, soil depth, the presence

or absence of waterlogged horizons, and drainage class were treated as fixed value per raster cell and soil depth, i.e., the mean of the DSM simulations was used ($SP_m$) (Table 1, Table 2). The presence of waterlogged soil horizons in the top soil layer (0-30 cm) was found for about 13 % of the case study area, for the 0-50 cm soil depth the figure was 27 %, and for the depth 0-100 cm, it was 40 % of the area. We assumed there was no waterlogging for the 0-10 cm depth because this was rarely observed in the data. About 74% of the agricultural soils were well drained (drainage class 1), 11% were moderately

well drained (class 2), and 15% were poorly drained (class 3)(Nussbaum et al., 2017b).

For the error propagation and the analysis of the uncertainty assessment results we distinguish two different types of SFA-methods depending on how the chosen random variables are taken into account in the calculation of the SFA methods (Figure 2). In cases where the SFA method consists of empirical equations (e.g., regression functions) or continuous PTFs, the variation of each $SP_d$ is fully propagated through these (type 1 equation). In our study this is the case for R-nutric, R-

carbon, and H-microorg. Some SFA methods such as R-water, R-nutril, R-acid, R-icont, H-plant, and P-agri are partly based on look-up tables using a classification of soil properties in the calculation, including PTFs that classify the estimation of secondary soil properties such as AWC (type 2 look-up tables). In particular, the method R-ocont classifies soil properties at the very beginning and groups the calculation of the retention of organic compounds in soils according to this classification.

**Type 1: equation**

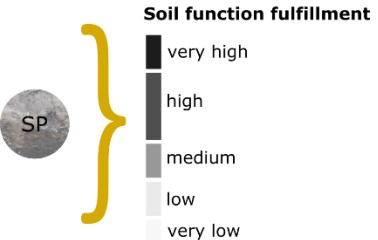

**Type 2: look-up tables**

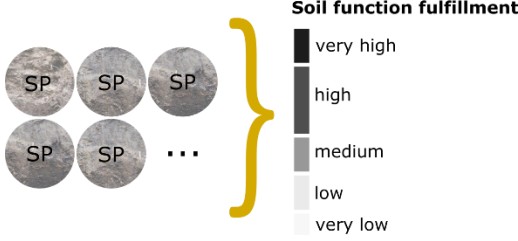

**Figure 2 Schematics of types of soil function assessment methods used in this study: a Type 1 equation directly rates a soil property (SP, possibly deduced by PTF, weighed or summed for a certain depth), a Type 2 look-up table combines two or more soil properties in a table to deduce SFF**



As a measure of uncertainty of the SFF scores for the ten SFA methods, we computed the interquartile range (IQR) for each raster cell, i.e., the difference between the 75% and 25% percentiles, and the ratio of IQR to the mean as an approximation for the coefficient of variation for the ordinal-scaled SFF scores. In order to visualize the uncertainty of the SFF scores in the soil function maps we generated two different map types. We visualized the uncertainty of the SFF scores resulting from the uncertainty of the four $SP_d$ values with the aim of facilitating communication in the decision-making process, and computed the probabilities < 10 %, 10-30% and >30% that the SFF score of a raster cell might deviate from the mean SFF score (only $SP_m$ used for SFA) for ± 1 or ± 2 or more SFF units. In this way, stakeholders might gain an overview of the areas of the case study area for which the SFF scores of individual soil functions have more or less confidence, expressed on the ordinal scale. The other type of maps allow visualization of SFF scores in a raster cell only where ≥90% of the 1000 simulated SFF scores were equal (C90), i.e., ≥90% of the simulated SFF scores revealed no variation indicating a high reliability of the result, whereas raster cells that do not meet this criteria are displayed as empty cells in the map. Additionally, 5% and 95% percentiles are displayed. As a measure of the overall uncertainty of a soil function, we calculated for each raster cell the median absolute deviation (MAD) and took the average of the MAD for all raster cells (MMAD). Finally, for more detailed analysis of the resulting uncertainty in the SFF scores for each assessed soil function, we computed the cumulative distribution functions (cdf) of the SFF scores including the mean of the deviations from the mean SFF score of a raster cell (MDM) for the 1000 simulations. The MDM was calculated separately for a) all simulations that were larger or b) smaller than the mean SFF score.

## 3    Results and Discussion
### 3.1    Mapping uncertainty of soil functions

Mapping the ten soil functions for the agricultural soils of the case study revealed pronounced spatial patterns, with a high variability of SFF scores across the region, linked to the inherent properties of the soils, terrain attributes, and climate conditions. The propagated uncertainties of soil properties $SP_d$ as produced by the SFA methods generally led to substantial uncertainty in the mapped soil functions, though to a different extent for individual soil functions and for subregions. Figure 3 presents the mean SFF scores for three selected soil functions and the associated uncertainties; the same maps for the other soil functions can be found in the Appendix. Figure 4 provides a general overview of the range of the SFF scores for the ten mapped soil functions and their uncertainties.

For instance, the regulation function for water (R-water) is in general higher for arable soils in the north-eastern part of the case study area, but is also associated with larger uncertainties. The water storage capacity (WSC) in our study area ranges between 44 mm and 270 mm (10% - 90% quantile, median: 204 mm) and the saturated hydraulic conductivity (SHC) ranges between 17 cm/d and 183 cm/d (median: 32 cm/d). The probability maps indicate that in the north-eastern part, 30% or more of the N= 1000 simulations did not fall in the "very high" SFF score, but scored one or two SFF categories lower, i.e., high or medium (Figure 3). Furthermore, the soils between Lakes Greifensee and Zürichsee in the western part of the region with


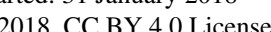


predominantly medium and low SFF scores were quite sensitive to uncertainties in soil properties. For the majority of soils in this subarea there is a relatively high probability that the mean SFF score for R-water might deviate by ± 1 SFF unit.

As expected, the calculation of the soil carbon pools was very sensitive to uncertainty in soil organic matter and stone content data (Figure 3, C-carbon). Carbon pools in agricultural soils are very heterogeneous across the case study area, with

5 low SFF scores mainly in the northern part ($< 10$ kg/m$^2$), with medium (13-15 kg/m$^2$) and high SFF scores (15-21 kg/m$^2$) in the southern part of the region. Mapping the associated uncertainty of soil carbon pools on an ordinal scale indicated, across almost the whole case study area, high probabilities that the SFF scores might deviate for ± 1 or even ± 2 SFF units. In contrast, the agricultural soils of the case study area showed high nutrient storage capacities throughout the region (Figure 3, R-nutric) and therefore, SFF scores of R-nutric were not that sensitive to the propagation of uncertainties of SP$_d$ through this

SFA method. Only in the north-eastern area did we observe some probabilities that SFF scores for R-nutric might be one SFF unit lower. Overall, the uncertainty of individual soil maps showed diverse spatial patterns, and mapping their uncertainty in the ordinal scale has the advantage that the communication of such uncertainties in decision-making in spatial planning processes improves levels of understanding, as proposed by (Bouma, 2014). Bouma (2014) stresses that this communication is a dialogue in which soil scientists should engage in order to link soil functions with ecosystem services, as

a means of connecting them to the demands and needs of stakeholders to find a balance in land-use planning between economic, social, and environmental aspects.

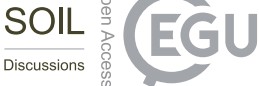

**Figure 3 Selected soil function maps for the agricultural land of the case study area and indication of their uncertainties in the ordinal scale: a) mean SFF scores (1st column) and b) probability that the mean SFF score of a raster cell deviates in the ordinal scale for ± 1 (2nd column) or c) ± 2 or more SFF units (3rd column) (raster cells 20 x 20 m², N=1000 simulations).**

5    The responses of the SFF scores for the assessed soil functions to uncertainty in the four simulated soil properties depend not only on the SFA method itself but also on the associated classification of the SFA results into the ordinal scale. In agreement





with the very high nutrient storage capacity of the soils, the basic soil properties of the grassland and arable soils are in a range that provides high and very high retention of trace metals (R-icont) as well, while the retention of organic chemical compounds in soil (R-ocont) is very low throughout the region (Figure 4) according to the assessment scale proposed in this SFA method (Litz 1998). Accordingly, the SFF scores for R-nutril, R-icont, and R-ocont are relatively insensitive to

uncertainty in soil properties, and the overall coefficient of variation is very small for these soil functions. The highest overall coefficient of variation was found for R-carbon and H-microorg, followed by R-acid and R-water (Figure 4). These results raise a question about the appropriate classification of SFA results from physical or chemical units into an ordinal assessment scale, and the adaption of such a classification for individual soil functions according to the range of soil properties for the case study area of interest or according to national references. Only where the SFF scores on the ordinal

scale of a certain soil function show substantial spatial variation can the influence of uncertain soil properties on the SFA results be investigated.

In this regard, H-plant is a special case for the assessment of uncertainties, because the outcome of this SFA method is a binomial variable, i.e., it indicates whether the soil provides niches conditions for rare plant populations or not. The simple SFA revealed that 14% of the soils in the case study area are suitable for providing niches for rare plants in terms of wet or

dry soil conditions, low nutrient availability and shallow soils. Such extreme soil conditions are mainly determined by soil depth, soil hydromorphic features, and other soil properties and only to some degree by the considered uncertainty of the soil properties $SP_d$. Therefore, for a proper uncertainty assessment of the SFA-method H-plant, not only must soil properties be taken into account, but the uncertainty of the aforementioned variables should also be considered.

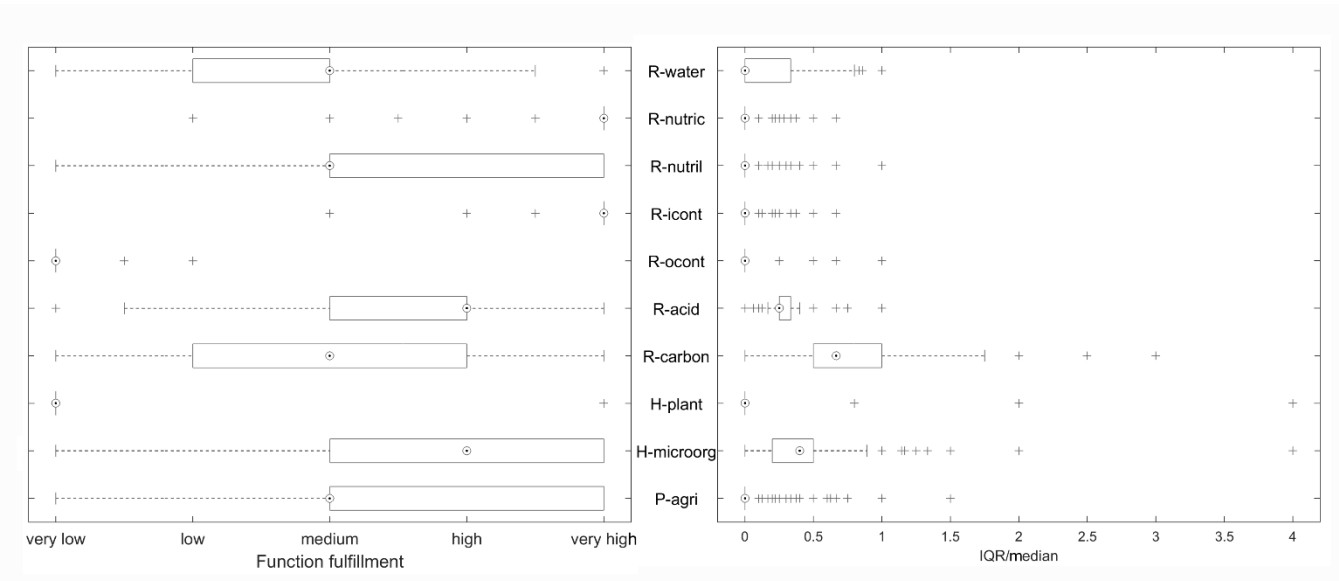

**Figure 4 General overview of the resulting range of SFF scores for the ten mapped soil functions (left), and of their coefficient of variation (right) expressed as the ratio of the interquartile range (IQR) and the median of the SFF scores for each raster cell. Circles with dots indicate the median coefficient of variation of the SFF scores across the case study area.**



In addition to the uncertainty maps described above, we generated supplemental information on the uncertainty of soil function maps addressing a given quality assurance criterion (Figure 5). We defined the C90 criteria, i.e., mean SFF scores for raster cells are displayed if at least 90% of the SFF score simulations result in the same SFF unit, otherwise the study area is shown as a grey area. In this way, stakeholders can easily gain an overview of those areas for which the soil function maps

5    are reasonably reliable. Figure 5 illustrates such supplemental maps and the visual effect of the C90 criteria for three SFA methods with high (R-nutril), medium (R-icont), and low reliability (H-microorg). Independent of the SFF scores, the number of raster cells displayed decreases for these three soil functions, in the same order. In sum, the uncertainty analysis shows that R-nutril and R-nutric fulfil the C90 criteria for most of the assessed agricultural area (85-90%); P-agri, R-water, R-icont, R-ocont fulfil them for about 41-51%; while R-acid, H-microorg and R-carbon apply for less than 5% of the case

10   study area. Accordingly, the average MAD of the SFF scores across the whole region increase noticeably for these three groups in the same order, from $< 0.01$ for the first group to $0.01 – 0.07$ for the second, and 0.43-0.88 for the third group. For the last group, the range of SFF scores (5% and 95% percentiles for each raster cell) in terms of SFF units varies for large areas from very low to very high, as illustrated for instance for H-microorg in the north-eastern part of the region (see Figure 5).







Figure 5 Uncertainty indication for soil function maps of R-nutril, R-icont and H-microorg: a) only mean SFF scores for raster cells are displayed if at least 90% of the N= 1000 simulations per raster cell revealed the same SFF score (first column). In addition, the range of SFF scores for each raster cell is shown: b) 5% and c) 95% percentiles of SFF scores, respectively (SFF = soil function fulfillment, grey: not C90 or no assessment, light grey: Lakes, "Arealstatisik"2009, 72 classes, © BFS 2010, GEOSTAT)





### 3.2    Cumulative distribution functions of SFF scores

Cumulative distribution functions (cdfs) of the SFF scores for all raster cells provided deeper insight into the sensitivity of

the SFA methods related to the uncertainty of the basic soil properties $SP_d$ with regard to the uncertainty for each SFF unit for each soil function. In general, we observed two different patterns in the cdfs of the SFF scores for type 1 (equation) and type 2 (look-up table) SFA methods (Figure 6 and 7).

For type 1 SFA-methods the uncertainty in the soil properties can be propagated entirely through regression functions and deterministic equations, and cdfs of the corresponding SFF scores indicate a smooth pattern of mean SFF scores and their

uncertainties from very low to very high SFF scores (Figure 6). In contrast, dependent on the classification of soil properties in the look-up tables used in type 2 SFA-methods, the cdf for R-nutril and P-agri show pronounced, and for P-water and P-acid less pronounced, step-functions for the mean SFF scores. Both of the first two SFA methods combine information on soils and environmental site conditions (e.g., geology, drainage systems, slope, altitude and climate) using various comprehensive look-up tables, leading to a strong discrimination of the final SFF scores for distinct ranges of soil properties.

Therefore, the outcomes of these SFA methods for a given region is not straightforward. For example, R-nutril combines texture, stone and soil organic matter content, bulk density, soil depth, drainage class, and environmental conditions as input data in various look-up tables. Thus, other input parameters including soil properties might also determine the main outcome of R-nutril for certain SFF units. For R-nutril and P-agri, soil depth and drainage class showed strong discrimination between SFF classes.

Figure 7 indicates that the SFF scores for R-nutril are only sensitive to some degree to the uncertainty in the soil properties $SP_d$ for high and very high SFF units, while for other SFF units other environmental data are dominant. Interestingly, we observe that certain SFF units of the type 2 SFA methods are more or less sensitive to the propagated uncertainty of soil properties $SP_d$ (Figure 7). This different response in the uncertainty of the SFF scores for the type 2 SFA methods was a priori unexpected and highlights the importance of such an uncertainty analysis of static SFA methods. The analysis provides

insight in terms of those SFF units for which uncertainty in soil property data plays an important role. For soils with a low suitability for food production the range of soil properties is not important (see Figure 7d) given that waterlogging or soil depth might be the dominant factors. However, for soils with medium and high suitability the range of soil organic matter, clay and stone content, and soil pH are decisive.

In line with the analysis of the uncertainty maps discussed above, relatively large uncertainty was found for all raster cells

for R-carbon and H-microorg (Figure 6). The SFA method H-microorg, for example, links microbial biomass for grassland and arable land use to soil organic matter, pH and clay content through an empirical PTF, and is therefore very sensitive to changes in soil properties. For R-water and P-agri for medium to very high SFF units the uncertainty in the soil properties $SP_d$ also leads to rather less confident SFF scores. Consequently, the analysis suggests that further measurements of basic




soil properties are required in the case study area to reduce the uncertainty in the spatial prediction of soil properties obtained from the DSM approach used by Nussbaum et al. (2017).

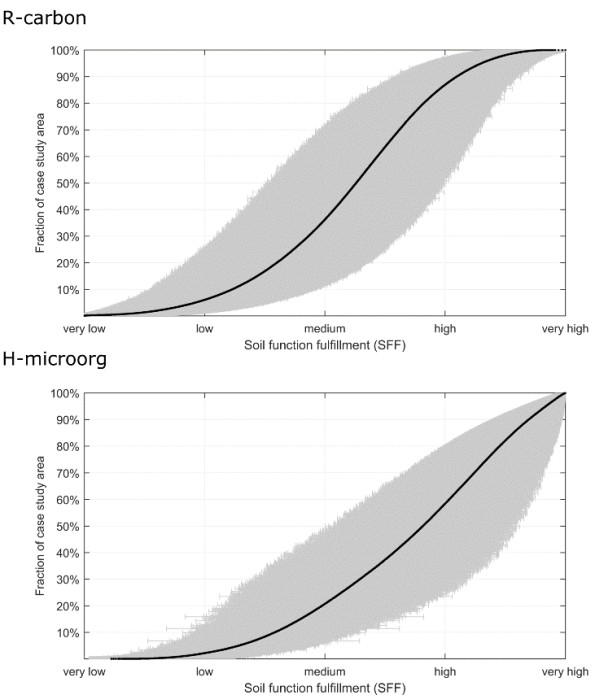

5   **Figure 6: Cumulative distribution function (cdf) of SFF scores for type 1 (equation) for   R-carbon and) H-microorg for agricultural soils of the case study area and the uncertainty resulting from four basic soil properties. (SFF score 1: very low to 5: very high; black: mean SFF score per raster cell, grey: range ± MDM per raster cell, number of raster cells: about 450 000; total area = 170 km² ).**



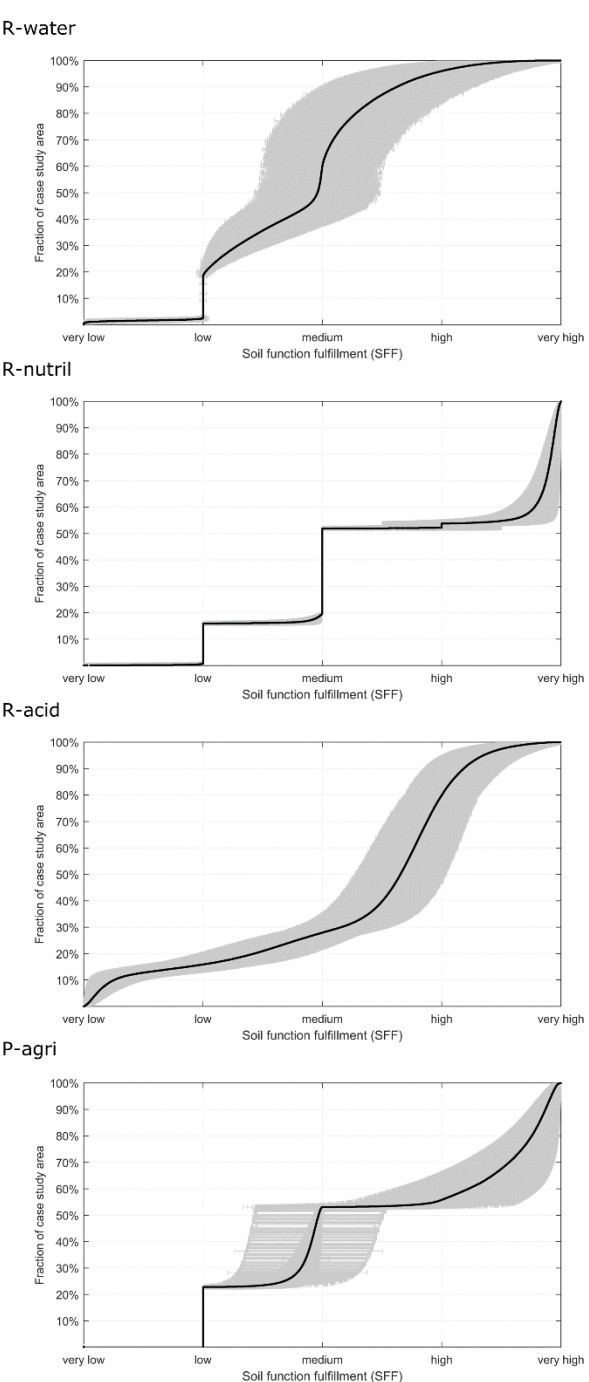

**Figure 7: Cumulative distribution function (cdf) of SFF scores for type 2 (look-up table) for R-water, R-nutril, R-acid and P-agri for agricultural soils of the case study area and the uncertainty resulting from four basic soil properties. (SFF score 1: very low to 5: very high; black: mean SFF score per raster cell, grey: range ± MDM per raster cell, number of raster cells for these soil functions ranged between 420 000-445 000; total area = 170 km$^2$).**



### 3.3    Uncertainty communication

Uncertainty is usually expressed as a probability of a state or an event, and can be presented numerically, verbally or graphically (IOM, 2013). Its presentation must fit the needs of the audience, the circumstances, and the purpose (IOM, 2013). We argue that the easiest way to interpret and the most suitable way of communicating (un-)certainties to actors in

land-use decisions is in the form of maps because this enables the visualization of spatial variability. Clearly, for a general overview of the study area, insight into method behaviour or comparisons between soil function, and information in the form of a table or a plot may also be suitable. In this study, we present readily communicable uncertainty indications for soil function maps. There are many other possibilities as well, of course, including statistically advanced methods to display (un)certainties in soil function maps. Rather than providing statistical measures, however, we advocate provision of simple

uncertainty maps such as those illustrated in Figures 3 and 5 as a means of facilitating the communication of uncertainties with stakeholders who may not be familiar with soil science and the contribution of soils to ecosystem services.

Experience of communicating uncertainty in the context of climate (Budescu, 2016) has shown that the use of simple phrases such as "very likely" combined with a numerical score (e.g., >90%) are of most value because stakeholders understand this kind of message the best. Communication of uncertainty through phrases has the advantage that they capture the attention of

stakeholders, although they are also somewhat open to individual interpretations in different contexts. According to (IOM, 2013), although graphical communications can "capture and hold people's attention", the interpretation may vary among individuals. One option would be to communicate a general phrase about the uncertainty of a soil function map, combined with a map that shows the details of the spatial variation of the uncertainty.

Depending on the method used, uncertainties in soil information input in SFA may be more or less disclosed or obvious, and

with this in mind the question itself is then what degree of uncertainty in data input in SFA should be transported through the SFA to match the needs of decision-makers in spatial planning processes. The optimal degree of uncertainty communication depends on the stakeholders involved in decision-making and the kind of decisions. The mindsets of the actors involved influence how the decision can profit from good quality soil function maps, including uncertainty indications. Time and resources for decision-making may vary and require a variable quality of information.


### 4    Conclusions

Decision-making in spatial planning processes should be well informed on the role of soils for society and the environment. Mapping of soil functions underpins the contribution of soils to ecosystem services, and is appropriate for communicating the importance of soils to spatial planners and other disciplines. In this study, we demonstrate how the reliability of soil

function maps can be assessed and communicated to allow for informed and transparent decisions in spatial planning processes, thereby helping to avoid poorly informed policy decisions with regard to available soil resources. Taking account of the uncertainty of basic soil properties, the performed uncertainty analysis for soil function assessment provides deeper insight into the sensitivity of soil function maps for the case study area. The cumulative distribution functions for the SFF scores of individual soil functions showed different patterns for SFA-methods based on empirical equations and SFA-

methods using simplified look-up tables. We propose two types of maps for the indication of uncertainties in soil function

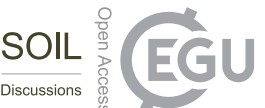

assessment, which supplement each other. We advocate that uncertainties in soil function maps should be made as transparent as possible and be visualized in easily understandable maps. Mainly because of computational limitations, we restricted our uncertainty analysis for predictive distributions to four soil properties at four depths. Other sources of uncertainty such as informational uncertainty on soil depth, soil hydromorphic features, and other environmental variables such as climate data and the reliability of PTFs should also be considered.

**Author contributions**

M. Nussbaum and A. Papritz developed and applied a DSM approach for our study area and provided soil property maps. L. Greiner implemented the soil function assessments and evaluated the results with contributions from M. Nussbaum, A. Gubler and A. Keller, who proposed forms of uncertainty indications in soil function assessments. With A. Keller, A. Papritz and M. Nussbaum, S. Zimmermann contributed to the conceptual and practical matching of soil properties and soil functions. A. Grêt-Regamey brought insights into spatial planning requirements. L. Greiner and A. Keller prepared the manuscript, to which all the co-authors contributed.

**Acknowledgements**

This study was part of the "Predictive mapping of soil properties for the evaluation of soil functions at regional scale" project (PMSoil; project number 406840-143096) and the "Matching soil functions and soil uses in space and time for sustainable spatial development and land management" project (OPSOL; project number 408640-143092), funded by the Swiss National Science Foundation within the framework of the National Research Programme "Sustainable Use of Soil as a Resource" (NRP 68), www.nrp68.ch. The Swiss Federal Office of the Environment (FOEN) provided additional funding.

**Conflicts of interest**

The authors declare that they have no conflict of interest.

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
