# Peer review of "Uncertainty indication in soil function maps –"

_SOIL, 2017_

## Referee Comment (RC1) · Anonymous Referee #1 · 3 Feb 2018

Manuscript soil-2017-41 Uncertainty indication in soil function maps –Transparent and easy-to-use information to support sustainable use of soil resources under review for journal SOIL is fitting this journal, and it is moderately stimulating. Tables: 2 Figures: 7 Appendix: 2 Citations: 56 (easily findable: 53; published after the year 2014: 9; SOIL: 4) Title: 19 words. Not very informative Abstract: not very informative Strengthens: a lot of data Weaknesses: aim, reproducibility

In detail (page.row): 1.20 SFA, please spell all the acronym out the first time they appear in the text 3.25 Please, describe the soils according to last edition of the WRB soil classification system (IUSS WG WRB, 2015). Main WRB Great Group probably deserve consideration at keyword level. 5.5 "The capacity of the soil to filter and buffer trace metals (R-icont) were assessed for cadmium, copper and zinc." A potentially misleading choice of elements in agricultural context. Please, explain why this choice. 9.5 Soil depth was treated as fixed value per raster cell. Quite strange choice in this geomorphological setting. Please, clarify maybe I did not understand well 11.5 The percentage of total variance attributed to internal variability and model uncertainty in the land carbon cycle comes normally mostly from model structure (e.g. DOI> 10.1126/science.aam8328) 10.21 "Mapping the ten soil functions for the agricultural soils." Soil functions include the production (agriculture) function. This definition is highly confusing.

The study present the results of primary scientific research. Experiments, statistics, and other analyses are performed to a sound technical standard and are described in moderate detail. Conclusions presented are unfocused, although supported by the data. The article is presented in a partially comprehensible manner. The research meets all applicable standards for the research integrity. The article does not adhere at all to appropriate reporting guidelines and community standards for data availability for replication purposes, the full raw experimental database must be available or deposited at relevant data repository (e.g. Zenodo). The research output, in terms of novelty, scores modest uniqueness, not introducing an original way of thinking. The level of clarity is partially good. The state of the art in literature is quite up-to-date. Figures are not all necessary and informative, for instance Figure 1 should be replaced by a kmz file and geographical coordinates clearly indicated. This paper does adopt a standard methodology in respect to the object of research. The main goal has been not accomplished as unclear. If the main goal is communication, the experiment should have been conducted (and then described) by measuring audience reactions. This is, actually, a methodology paper. But, the Authors must clearly explain what the innovative part of the proposed method is. The paper does not fully discuss the limitations of the approach and potential biases due to the assumptions made. Moderate is forecast its potential impact upon the international scientific community of reference. Refer-

ences IUSS Working Group WRB. 2015. World Reference Base for Soil Resources 2014, update 2015 International soil classification system for naming soils and creating legends for soil maps. World Soil Resources Reports No. 106. FAO, Rome IT, 192 p.

---

## Referee Comment (RC2) · W. Towers (Referee) · 20 Feb 2018

I enjoyed reading this paper and it revives memories of a similar project in which I was involved in the mid 2000s; it is good to see how the science has progressed since then. It represents the sort of research that demonstrates the value of soils science and data to wider society. that I have swayed between minor and major revision and I will leave it to the editor to decide whether my principal comments need to be addressed.

I attach the paper with some minor comments and suggestions in 'sticky notes' in addition to those below. Please let me know if it is not attached, it appeared to attach very quickly!

[Figure]

The paper relies heavily on the output from the DSM exercise which models data from 418 data points in the study area (170 square kilometres). From my experience this study area is 'data heavy' - over 2 observations to a metre depth per square kilometre. Are these data from a specific grid survey or does it represent the density of observations across Switzerland? This poses the question of whether this approach can be replicated across larger areas to the same degree of detail. The DSM appears to have been conducted independently of soil type; what was the reason (s) for this?

The study area is a curious shape. It would have been beneficial if it had been a river catchment or an administrative area and was it chosen, at least in part, because of data availability?

The DSM was carried out using soil legacy data for which no detail is provided . The paper would benefit from some information on the age, purpose and the attributes within these data; it would make the paper more transparent and the reader would understand the opportunities and limitations of such data. Are they still fit for purpose? Don't worry,it is an issue for soil science everywhere!

Use the term 'sub-functions' throughout, they are not the high level functions.

Some figures are very good e.g. Figures 4, 6 and 7 whereas others are too small (3, 5) and do not encourage scrutiny and I would suggest screening out the areas that are not assessed (mainly forestry) on Figure 1.

Table 1 stretches across two pages and much of the title would be better as a footnote below it.

I appreciate the problem but some sections are quite difficult to read as they are 'acronym heavy' e.g. Section 2.4. The use of acronyms for the models and their outputs add to this but I cannot suggest a better way I'm afraid.

Please also note the supplement to this comment:

https://www.soil-discuss.net/soil-2017-41/soil-2017-41-RC2-supplement.pdf

[Figure]

**Supplement:**

[revised manuscript text omitted]

---

## Author Comment (AC1) · 23 Feb 2018

We thank the referee for his/her feedback and the helpful comments. We refer to the comments below and propose adjustments to the manuscript, see supplement.

RC = referee comment

AR = authors response

P = page

L = line

[Figure]

General Feedback

Comment 1

RC: Not very informative abstract.

AR: We revised the abstract and provide more information with regard to the context of our study, see manuscript in the supplement.

Comment 2

RC: Conclusions presented are unfocused, although supported by the data.

AR: We have to agree and tried to provide a better structure in the conclusions by being more focused on the objectives of the study presented in the introduction, see supplement. Further, we tried to prevent a possible misunderstanding by renaming chapter 3.3 "Uncertainty communication" to "Thoughts on uncertainty indication" and eliminating the term "communication" from the conclusions. We renamed chapter 3.3., also in view of your comment about the unclear goal of our paper (comment number 4). We formulated two goals, chapters 3.1. and 3.2. answer to these two goals, chapter 3.3. shortly discusses more general thoughts on the topic of uncertainty indications.

Comment 3

RC: Figures are not all necessary and informative, for instance Figure 1 should be replaced by a kmz file and geographical coordinates clearly indicated.

AR: We include the kmz-file in the supplement and added the coordinates in the legend of Figure 1, see manuscript in the supplement. We presume that that Figure 1 helps the reader to understand the subsequent figures presenting the soil function maps. We agree that Figure 2 is not really necessary and eliminated that Figure.

Comment 4

RC: The main goal has been not accomplished as unclear. If the main goal is communication, the experiment should have been conducted (and then described) by measuring audience reactions.

AR: For our study, we formulated two objectives: 1) to indicate uncertainty in soil function maps that are caused by informational uncertainty and spatial variation of soil properties and 2) to show how sensitive the chosen SFA methods are to error propagation. The revised abstract and conclusion sections should be now more informative and providing in a better way the context of our objectives. Moreover, we used the general term "communication" in chapter 3.3 to address stakeholder demands. We are now more precise and avoid that general term (see as well 2nd comment). Communication is an important aspect in assessing soil functions though, as the aim of soil function assessment is to simplify the medium "soil" and provide information in an easy-to-understand (and thereby easy-to-communicate) form for non-soil scientists. We therefore adapted and slightly expanded the discussion-chapter 3.1. "Mapping uncertainty of soil functions" also naming possible advantages of our approach, see supplement, in order to provide more context to our goal and clarify the aspect of communication. Furthermore, we added in chapter 3.2. "Cumulative distribution functions of SFF scores" a short discussion section providing more details.

Comment 5

RC: This is, actually, a methodology paper. But, the Authors must clearly explain what the innovative part of the proposed method is.

AR: To our knowledge, uncertainty propagation in static soil function assessment has not been performed yet. In addition, we are not aware of any recommendations how to visualize uncertainties in soil function maps. So, the methods applied are not new and are standard in other science disciplines but have not been applied for the assessment of soil functions. We refer to that aspect in the introduction chapter (P3L14).

Comment 6

RC: The paper does not fully discuss the limitations of the approach and potential biases due to the assumptions made.

AR: We chose the topic to foster information on accuracy of soil function maps in spatial planning programs. By covering a set of 10 soil functions, we tried to consider a broad part of soil multifunctionality and each soil function assessment method contains limitation. Also, uncertainty indication is restricted to four soil properties used in soil function assessment. We tried to clarify these main limitations in the conclusions-chapter, see supplement.

Comment 7

RC: The article does not adhere at all to appropriate reporting guidelines and community standards for data availability for replication purposes, the full raw experimental database must be available or deposited to relevant data repository (e.g. Zenodo). / a lot of data Weaknesses: reproducibility

AR: We used soil property predictions provided by Nussbaum et al. (2017). In the study of Nussbaum et al. (2017) soil data were used under a non-public data licence (Canton of Zurich, contract number TID 22742; WSL). We refer now to the data availability in on P8L1.

Line-specific Feedback

P1L20

RC: SFA, please spell all the acronym out the first time they appear in the text

AR: Corrected.

P3L25

RC: Please, describe the soils according to last edition of the WRB soil classification system (IUSS WG WRB, 2015). Main WRB Great Group probably deserve consideration at keyword level.

AR: We used the last WRB edition (IUSS Working Group WRB, 2015) and agree on the degree of detail: we included the most common principal qualifiers per Reference Soil Group for our study area in the manuscript P3L31, see supplement

P5L5

RC: "The capacity of the soil to filter and buffer trace metals (R-icont) were assessed for cadmium, copper and zinc." A potentially misleading choice of elements in agricultural context. Please, explain why this choice.

AR: We added reasons why these trace elements are relevant for arable and grassland soils addressing the fertilizer types containing these elements. Furthermore, we provide now references to some relevant studies to underpin the choice of the three trace elements, see manuscript P5L17.

P9L5

RC: Soil depth was treated as fixed value per raster cell. Quite strange choice in this geomorphological setting. Please, clarify maybe I did not understand well

AR: This was probably formulated in a misleading way. We did not use soil depth as a random variable and used the mean predictive value (SPm) for soil depth per raster point instead. To be clear, we eliminated the formulation "treated as fixed value" from P9L10.

P11L5

RC: The percentage of total variance attributed to internal variability and model uncertainty in the land carbon cycle comes normally mostly from model structure (e.g.DOI> 10.1126/science.aam8328) AR: We did not model C-pools but simply calculated C-pool to 1m or soil depth from data on soil organic matter, stone content, bulk density obtained from DSM by Nussbaum et al. (2017).

P10L21

RC: "Mapping the ten soil functions for the agricultural soils." Soil functions include the production (agriculture) function. This definition is highly confusing.

AR: We meant to stress that we only assessed soils under agricultural use and excluded soils under forests, in settlements, parks etc.. We adapted the sentence to "Mapping the ten soil functions for the study area..." to avoid this double meaning.

References

Nussbaum, M., Walthert, L., Fraefel, M., Greiner, L. and Papritz, A.: Mapping of soil properties at high resolution in Switzerland using boosted geoadditive models, SOIL , 2017, 1–32, doi:10.5194/soil-2017-13, 2017b.

IUSS Working Group WRB. 2015. World Reference Base for Soil Resources 2014, update 2015 International soil classification system for naming soils and creating legends for soil maps. World Soil Resources Reports No. 106. FAO, Rome.

Please also note the supplement to this comment:
https://www.soil-discuss.net/soil-2017-41/soil-2017-41-AC1-supplement.zip

---

## Author Comment (AC2) · 13 Mar 2018

**Response to**
**Interactive comment by W. Towers on "Uncertainty indication in soil function maps - Transparent and easy-to-use information to support sustainable use of soil resources" by Lucie Greiner et al.**

We thank Willie Towers for his valuable feedback, we refer to the comments below and to adjustments in the manuscript.

[Figure]

*RC = referee comment*
AR = authors response
P = page
L = line

**General Feedback**

Comment 1
*RC: The paper relies heavily on the output from the DSM exercise which models data from 418*
*data points in the study area (170 square kilometres). From my experience this study area is*
*data heavy - over 2 observations to a metre depth per square kilometre. Are these data from a*
*specific grid survey or does it represent the density of observations across Switzerland? This*
*poses the question of whether this approach can be replicated across larger areas to the same*
*degree of detail.*
AR: The DSM approach is described in detail in Nussbaum et al. (2017).
Soil organic matter for 50-100 cm depth was available for 418 data points in our study area.
We tried to clarify this sentence, see supplement (manuscript P9L11). For other soil properties
there were more data points. The region is indeed data heavy and the approach cannot be
replicated to larger areas to the same degree of detail. We propose to include this fact in the
limitations-section in the conclusions, see manuscript P21L23 ff. in the supplement.

Comment 2
*RC: The DSM appears to have been conducted independently of soil type; what was the*
*reason (s) for this?*
AR:Soil function assessment methods are based on soil properties and Nussbaum et al.
(2017) aimed at generating soil property maps. We now emphasize this fact in the manuscript
P8L8, see supplement.

Comment 3
*RC: The study area is a curious shape. It would have been beneficial if it had been a river catchment or an administrative area and was it chosen, at least in part, because of data availability?*
AR: Several project partners within the Swiss National Research Programme "Soil as a resource" (www.nrp68.ch) worked in the presented study area. The curious extent of the study area is due to criteria, which had to be met covering the needs of the project partners. Beside other criteria, the area had to be covered by APEX Swiss Earth Observatory Network (www.seon.uzh.ch) flights, which gathered spectroscopic data. We shortly explain the extent in the manuscript P4L5 ff., see supplement.

Comment 4
*RC: The DSM was carried out using soil legacy data for which no detail is provided. The paper would benefit from some information on the age, purpose and the attributes within these data; it would make the paper more transparent and the reader would understand the opportunities and limitations of such data. Are they still fit for purpose? Don't worry, it is an issue for soil science everywhere!*
AR: We agree and provide some detail in the manuscript P8L15 ff., see supplement and refer to Nussbaum et al. (2017) for more detail.

Comment 5
RC: Use the term 'sub-functions' throughout, they are not the high level functions.
AR: Corrected, see supplement.

Comment 6
*RC: Some figures are very good e.g. Figures 4, 6 and 7 whereas others are too small (3, 5) and do not encourage scrutiny and I would suggest screening out the areas that are not assessed (mainly forestry) on Figure 1.*
AR: We enlarged Figures 3 and 5, see supplement, and would want to visualize settlements and forests in the study area, even though not assessed, for a better overview in Figure 1.
Comment 7
*RC: Table 1 stretches across two pages and much of the title would be better as a footnote below it.*
AR: Changed.

Comment 8
*RC: I appreciate the problem but some sections are quite difficult to read as they are 'acronym heavy' e.g. Section 2.4. The use of acronyms for the models and their outputs add to this but I cannot suggest a better way I'm afraid.*
AR: We agree, used descriptions instead of acronyms only, structured parts of section 2.4. and hope, this facilitates reading, see manuscript P10L10 ff.

**Feedback in manuscript**

P1L23
*RC: Soil functional assessment? This needs spelled out as many readers might only read the abstract.*
AR: Corrected, see supplement.

P2L12
*RC: Reference should also be made EU Soil Thematic Strategy to demonstrate their policy and societal relevance.*
AR: We agree and included the reference in the manuscript P2L16 ff., see supplement.

P3L22
*RC: Figure 1?*
AR: Corrected.

P3L26

*RC: variable*
AR: Corrected.

P3L28
*RC: growing season?*
AR: Corrected.

P3L29
*RC: The study area is a very contrived shape; has it been chosen because of data availability? A catchment or administrative area would have been more appropriate.*
AR: See response to comment 3.

P6L12
*RC: It is a little unusual to refer forward in a paper.*
AR: That is true. We deleted the reference. It could be more confusing than helpful and it is not necessary to understand Table 1.

P9L4
*RC: Does this really involve that much computing capacity? 4 variables in 418 data points.*
AR: See response to comment 1, we tried to clarify the misunderstanding of 418 data points and included the number of raster cells in our study area in the manuscript at P9L17, see supplement.

P10L17
*RC: This paragraph is quite difficult to follow and 'digest'. Indeed the paper has a lot of acronyms throughout.*
AR: See response to comment 8.

**Reference**

[revised manuscript text omitted]

20  static SFA, in order to 1) indicate how accurate the SFA results are in response to informational uncertainty and spatial variation of soil properties as quantified by the DSM approach, and 2) to gauge how sensitive the SFA methods are to predictive distribution in soil properties.

**2    Materials and Methods**

**2.1    Study area**

Our study area is located in the Swiss Plateau in the Canton of Zürich around Lake Greifensee, see Figure 1. The

30  region is dominated by urban areas and agricultural land (crop production, mixed and dairy farming). We only assessed soils under agricultural use. Urban areas, forest, wetlands, parks, and city gardens are excluded from this study, resulting in a total study area of 170 km$^2$.  Chromic, Calcaric and Eutric Cambisols (63% of study area), Stagnic, Reductigleyic and Calcaric Gleysols (20% of study area), Haptic Luvisols (11% of study area) and Hemic, Drainic Histosol

and Calcaric , Eutric Fluvisols or Regosols,  have developed in a variable geology, but in general on quaternary molasses and moraines. The region lies at about 390-840 metres above sea level, and the growing season amounts to approximately 190 days per year. Slopes greater than 35% can only be found alongside moraines, otherwise the slopes are between 10 and 15% (Jäggli et al., 1998). The shape of the study area is formed by administrative boundaries in the south east and otherwise by APEX spectroscopy flight bands (www.seon.uzh.ch). More details on the region, its soils and its extent are provided in Jäggli et al., (1998) and Nussbaum et al. (2017b).

[Figure]

**Figure 1 Study area in the Swiss Midlands, 672489 - 715769 X, 228156 – 259960 Y, GCS_CH1903. (Orthophotos study area: SWISSIMAGE 2005, ©SWISSTOPO. Administrative boundaries Europe: NUTS 2010, ©EuroGeographics)**

**2.2    Soil function assessment**

We assessed regulation, habitat and production functions for 10 soil (sub-)functions (Table 1) as proposed in a previous review by Greiner et al (2017). Each SFA method addresses a certain domain of the soils multi-functionality depicting a

specific assessment criterion, e.g., the nutrient storage capacity of soils for the nutrient cycle. The SFA methods require data on soil properties, PTFs, and other environmental data (Table 1).

**2.2.1 Regulation functions**

5 We assessed the regulation of the water cycle (R-water) following the method proposed by (Danner et al., 2003), which combines the water storage capacity (WSC in mm/m$^2$) of soils with their saturated hydraulic conductivity (SHC in cm/day) for a reference soil depth down to 1m. The nutrient storage capacity (NSC in mol$_c$/m$^2$) of soil is one of its most important parameters, determining the nutrient cycle (R-nutric). We calculated the NSC according to (Lehmann et al., 2013), multiplying the fine earth fraction (mass of clay and silt) and the amount of soil organic matter for each soil layer with its

10 effective cation exchange capacity (CEC$_{eff}$) down to a soil depth of 1 m. The method proposed by (Jäggli et al., 1998) evaluates the capacity of soils to prevent the loss of soil nutrients by runoff and percolation to ground and surface water (R-nutril). The SFA method takes into account basic soil properties as well as the hydromorphic properties of soils (waterlogging) and environmental site conditions. The capacity of the soil to filter and buffer trace metals (R-icont) were assessed for cadmium, copper and zinc using a method developed by the German Association of Water, Wastewater and

15 Waste (DVWK, 1988) to prevent groundwater pollution by trace elements. The SFA method evaluates the filtering capacity of topsoils (0- 30 cm) to retain trace metal cations based on sorption sites of organic matter, clay minerals, and sesquioxides in conjunction with soil pH and redox potential (DVWK, 1988). Agricultural soils are potentially treated with commercial fertilizers, animal manure, compost, waste-derived fertilizers, and pesticides, which contain nutrients and trace metals such as cadmium, copper and zinc. While copper mainly stems from fertilizers, cadmium and zinc are brought into soils by

20 manure as well. Additionally, farmers may use pesticides containing zinc and copper (Jensen et al., 2016; Six and Smolders, 2014; Keller and Schulin, 2003):

[revised manuscript text omitted]

For the error propagation and the analysis of the uncertainty assessment results we distinguish two different types of SFA-methods depending on how the chosen random variables are taken into account in the calculation of the SFA methods . In cases where the SFA method consists of empirical equations (e.g., regression functions) or continuous PTFs, the variation of each soil property with probability distribution, $SP_d$, is fully propagated through these (type 1 equation). In our study this is the case for methods assessing regulation of nutrient cycle, carbon cycle and habitat for microorganisms (R-nutric, R-carbon, and H-microorg).  SFA methods assessing soils regulation of water cycle, nutrient losses, acidification, inorganic contaminants, habitat for plants or agricultural production function (R-water, R-nutril, R-acid,

R-icont, H-plant, and P-agri) are partly based on look-up tables using a classification of soil properties in the calculation, including PTFs that classify the estimation of secondary soil properties such as  available water capacity (type 2 look-up tables). In particular, the method assessing soils regulation of organic contaminants (R-ocont) classifies soil properties at the very beginning and groups the calculation of the retention of organic compounds in soils according to this classification.

**Type 1: equation**

[Figure]

**Type 2: look-up tables**

[Figure]

We computed a) two measures of uncertainty for SFF scores, b) two types of maps visualizing uncertainties, c) two measures for overall uncertainty per soil sub-function in our study area and show d) uncertainties of SFF scores per soil sub-function in detail.

[revised manuscript text omitted]

5  advantage that the communication the common understanding of spatially heterogenic uncertainties in SFF of such uncertainties between actors in decision-making in spatial planning processes improves levels of understanding. This adds information on reliability for the , to the well suited soil function maps used to communicate the value of soils values to spatial planners and other disciplines (Haslmayr et al., 2016; Sanchez et al., 2009), thus supporting allowing for a bettermore confidence in accurate land use decisions. Moreover, revealing the reliability of soil function maps transparent,

10 and thereby higher quality soil function maps mightcan also support efforts to strengthen the link between soil functions and ecosystem services. This link is important, as proposed by (Bouma, 2014). Bouma (2014) stresses: that this communication is a dialogue in which soil scientists should engage in order to link soil functions with ecosystem services are, as a means of connecting them soil functions to the demands and needs of stakeholders to find a balance in land-use planning between economic, social, and environmental aspects, a balance crucial to find (e.g.,Bouma 2014; Grêt-Regamey et al., 2017;

15 Valujeva et al. 2016).

[revised manuscript text omitted]

Moreover, our analysis clearly indicates that SFA results are not comparable between type 1 and type 2 methods and among

5   type 2 methods in view of uncertainty indication. One of the core aspects of the soil function concept is to assess soils multifunctionality and the role soils play for humans and the environment in general and to support land use decisions (e.g., Haygarth and Ritz, 2009; Schulte et al., 2014). In this study, we argue that the weighing between the importance of different soil functions and other goods should be the result of a regional or political valuation process. The valuation of soil is more straightforward, if SFF scores are comparable and retain a comparable amount of uncertainty – even though soil functions

10  are not or only weakly comparable of course. This ordinal comparability allows to deliberate on the importance of soil functions via SFF scores and soil function maps. Deliberation is seen as a promising tool to value environmental goods or services (Vatn, 2009). Soil function maps including uncertainty indications can also be used in multi-criteria decision analysis (MCDA), for an example in spatial planning including soil, see Grêt-Regamey et al. (2017). In terms of ecosystem services-language: soil function maps show the supply of a soil contribution to ecosystem service, the demand of these

15  services has to be assessed on another level. Still, the supply-information should meet the needs of the further processes.

[revised manuscript text omitted]

35    quality of the information used for decision-making. In this study, we try to foster transparency in two ways.

1) We demonstrate how the reliability of soil function maps can be  presented to allow for informed and transparent decisions in spatial planning processes, thereby helping to avoid poorly informed policy decisions with regard to available soil resources. We propose two types of maps for the indication of uncertainties in SFA, which supplement each other. We advocate that uncertainties  should be made as transparent as possible and be visualized in easily understandable maps.

2) Taking account of the uncertainty of basic soil properties, the performed uncertainty analysis for SFA provides deeper insight into the sensitivity of the SFA methods for the uncertainties of four soil properties in our case study area. The cumulative distribution functions for the SFF scores of individual soil functions showed different patterns for SFA-methods based on empirical equations and SFA-methods using simplified look-up tables.

The main limitations of this study are clear: We restricted  uncertainty  propagation in SFA  to predictive distributions  of four soil properties at four depths out of eight soil properties used for SFA, mainly because of computational limitations. Other sources of uncertainty such as informational uncertainty of  other environmental variables such as climate data and the reliability of PTFs should also be considered. Further, we used one SFA method per soil function and besides this fact one can challenge the methods: for example on the soil data used, the inclusion of other environmental data, the use of PTFs, the assessment depth, the ordinal assessment scale, the calibration of the assessment scales, the simplifications made to depict static soil function fulfillment. The SFA approach in general is flexible and modular, methods can be adapted or exchanged, but an issue of the approach is that validation of an assessment result is hardly possible (Calzolari et al., 2016). Although we used established SFA methods, we still consider the development of SFA methods an ongoing task and hope to contribute by showing that the choice of method matters in view of uncertainty propagation. Additionally, soil data availability for the study area was good in comparison to other areas in Switzerland. To achieve the same degree of detail in applying this approach for larger areas without soil sampling could therefore be challenging.

**Author contributions**

[revised manuscript text omitted]

Six, L.; Smolders, E. Future trends in soil cadmium concentration under current cadmium fluxes to European agricultural

5  soils. Sci. Total Environ., 485–486, 319–328, doi: 10.1016/j.scitotenv.2014.03.109, 2014.

Swisstopo: Geologische Karte der Schweiz 1:500'000, 2017(18.05.2017) [online] Available from: https://shop.swisstopo.admin.ch/de/products/maps/geology/GK500/GK500_DIGITAL, 2008.

10  Swisstopo: SwissAlti3d, edited by Federal Office of Topography, 2014.

Taylor, A. L., Dessai, S. and de Bruin, W. B.: Communicating uncertainty in seasonal and interannual climate forecasts in Europe, Philos. Trans. R. Soc. A Math.Physical Eng. Sci., 373(2055), doi:10.1098/rsta.2014.0454, 2015.

Tóth, G., Gardi, C., Bódis, K., Ivits, E., Aksoy, E., Jones, A., Jeffery, S., Petursdottir, T. and Montanarella, L.: Continental-scale assessment of provisioning soil functions in Europe, Ecol. Process., 2(32), 2013.

15  Valujeva, K., O'Sullivan, L., Gutzler, C., Fealy, R., Schulte, R.P.O.: The challenge of managing soil functions at multiple scales: An optimisation study of the synergistic and antagonistic trade-offs between soil functions in Ireland. Land use policy 58, 335–347, 2016. https://doi.org/https://doi.org/10.1016/j.landusepol.2016.07.028

Vatn, A.,: An institutional analysis of methods for environmental appraisal. Ecological Economics, 8(68), 2009. https://doi.org/10.1016/j.ecolecon.2009.04.005

20  van der Sluijs, J. P.: Uncertainty and precaution in environmental management: Insights from the UPEM conference, Environ. Model. Softw., 22(5), 590–598, doi:10.1016/j.envsoft.2005.12.020, 2007.

Vaysse, K. and Lagacherie, P.: Evaluating Digital Soil Mapping approaches for mapping GlobalSoilMap soil properties from legacy data in Languedoc-Roussillon (France), Geoderma Reg., 4, 20–30, doi:10.1016/J.GEODRS.2014.11.003, 2015.

25  Vereecken, H., Schnepf, A., Hopmans, J. W., Javaux, M., Or, D., Roose, T., Vanderborght, J., Young, M. H., Amelung, W., Aitkenhead, M., Allison, S. D., Assouline, S., Baveye, P., Berli, M., Brüggemann, N., Finke, P., Flury, M., Gaiser, T., Govers, G., Ghezzehei, T., Hallett, P., Hendricks Franssen, H. J., Heppell, J., Horn, R., Huisman, J. A., Jacques, D., Jonard, F., Kollet, S., Lafolie, F., Lamorski, K., Leitner, D., McBratney, A., Minasny, B., Montzka, C., Nowak, W., Pachepsky, Y., Padarian, J., Romano, N., Roth, K., Rothfuss, Y., Rowe, E. C., Schwen, A., Šimůnek, J., Tiktak, A., Van Dam, J., van der

30  Zee, S. E. A. T. M., Vogel, H. J., Vrugt, J. A., Wöhling, T. and Young, I. M.: Modeling Soil Processes: Review, Key Challenges, and New Perspectives, Vadose Zo. J., 15(5), 0, doi:10.2136/vzj2015.09.0131, 2016.

Walthert, L., Bridler, L., Keller, A., Lussi, M., Grob, U. Harmonisierung von Bodendaten. Anhang zum Schlussbericht von PMSoil ("Predictive mapping of soil properties for the evaluation of soil functions at regional scale", NRP 68).

Eidgenössische Forschungsanstalt WSL und Agroscope Reckenholz. ETH Zurich Research Collection. https://doi.org/https://doi.org/10.3929/ethz-a-010801994, 2016.